# Study on the Status of Gamete Freezing and Reproductive Health of Korean Adults Aged 19–49 Years

**DOI:** 10.3390/healthcare13030210

**Published:** 2025-01-21

**Authors:** Jiyoung Song, Eunwon Lee

**Affiliations:** 1Department of Nursing, Hoseo University, Asan 31499, Republic of Korea; 2Department of Nursing, Doowon Technical University, Ansung 17520, Republic of Korea

**Keywords:** gamete freezing, reproductive health, adults, fertility preservation

## Abstract

Background: With advances in biomedical technology and social changes, such as a high rate of late marriages, the interest in gamete freezing is increasing. Therefore, this study aimed to investigate the status of gamete freezing and reproductive health of adult Koreans aged 19–49 years using data from the 2021 National Family and Fertility survey. Method: The SPSS program was used to analyze the data using descriptive statistics, Fisher’s Exact *p*-value, and the *t*-test. Results: Of the 14,040 study subjects, 101 (0.7%) had frozen their gametes. The average age of this group was 38.1 years, 61.4% were women and 38.6% were men. In addition, 83.2% of the group had a university degree or higher, 85.1% lived in urban areas, 94.1% were married, and 66.3% were economically active. The participant-reported incidence of genital infections in this group was 29.7%. The obstetric history revealed that 91.9% of the women considered themselves infertile. Conclusions: The results of this study can be used as basic data for developing educational programs to preserve fertility or developing childbirth-related policies in the future.

## 1. Introduction

The advances in technology, such as cryopreservation, as well as the preimplantation genetic diagnosis of embryos, have significantly aided successful conception using assisted reproductive techniques in couples with infertility [1]. Clinical pregnancy and newborn birth rates using the cryopreservation of gametes through vitrification have been found to be similar to that observed with fresh gametes, with no differences observed in the incidence and type of complications related to childbirth or the health of newborns [2]. In Korea, infertility treatment costs are covered by health insurance, and the accessibility to in vitro fertilization has improved [3]. People’s interest in cryopreservation for various reasons, including their changing social needs, has steadily increased in the last few years, and the number of stored frozen oocytes has risen about 2.5-fold from 44,122 in 2020 to 105,523 in 2023 [4].

This phenomenon has increased the interest in oocyte cryopreservation for non-medical reasons, not only to preserve fertility in female patients with incurable diseases but also among highly educated women, enabling them to make life choices on when they would like to conceive and enhance their reproductive potential [5,6]. Egg-freezing technology was first introduced in the mid-1980s to preserve the fertility of cancer patients who may be affected by the detrimental effects of chemo- and radiation therapy on fertility. However, it is now gaining attention as a procedure for women who get married late or are opting for late pregnancies [7,8]. In the case of men, sperm freezing began in individuals with incurable diseases, those in military service, and those who underwent a transition. However, it has recently been performed for non-medical reasons to extend the childbearing phase in an individual’s life [9,10]. This is related to the changing reality in which there is an increase in demand to preserve fertility due to concerns of infertility caused by aging (as a result of late marriages) and environmental pollution, and the social understanding and acceptance of the same [8,11].

Pregnancy and childbirth, which are natural processes, are now being artificially performed in the laboratory at human will [12]. However, in addition to issues associated with the technology itself, the various socio-economic and ethical aspects should also be considered [8,13]. This is because social, ethical, and legal problems may arise during gamete collection and freezing, embryo creation, and non-medical freezing [3,14,15]. Gamete freezing, including the initial freezing procedure, which typically covers the cryopreservation of embryos and the first year of storage is expensive. Hence, only a few people can afford it and this can deepen social inequality [15,16]. In this sense, the government and related organizations should establish appropriate measures to manage gamete collection and freezing so that the social order is not disturbed or conflicts are not caused [1,17].

The success rate of pregnancy after gamete freezing varies depending on the woman’s age, number of fertilized eggs, and other factors, and it is known that older women are at a higher risk of early fetal loss due to the impact on chromosomal and genetic disorders [16]. However, studies [1,7] indicate that people who want to consider gamete freezing lack the related knowledge. In addition, people experience a lot of anxiety because of a lack of awareness regarding the gamete freezing process, and there are concerns about the social stigma associated with it, as well as the result of the procedure [16,18]. Considering the increasing population of infertile men and women and the resultant low birth rate, it is necessary to provide accurate information related to gamete freezing to present it as a specific alternative for preserving fertility for the future and give unmarried men and women a chance to choose this process [7,8,19].

Previous studies on gamete freezing have mainly focused on assessing the awareness of the issues related to freezing [1,6] or the technical aspects of the gamete freezing process [20]. In terms of research subjects, many studies have been conducted on patients with serious diseases such as breast cancer [21,22] or individuals with infertility [17]. There are few studies on the status of gamete freezing including representative individuals of the general population. Such studies are necessary because the implementation of assisted reproductive technology and gamete freezing have various social, ethical, and legal problems [14] due to the ultimate goal of human birth [2,15]. Therefore, this study investigated the status of gamete freezing in the population covered by the 2021 National Family and Fertility Survey conducted by the Korea Institute for Health and Social Affairs [23]. The results can be used as basic data for educating individuals or policy development.

## 2. Materials and Methods

### 2.1. Study Participants

This study was based on the data of 14,538 Korean adults aged 19 to 49 years who participated in the 2021 National Family and Fertility Survey to understand the status of gamete storage and reproductive health among them. The survey is conducted every three years on adults and their spouses to observe changes in the life course and family path of individuals through demographic behavior, such as marriage and childbirth. The data were collected by a trained investigator who visited the sample household, explained the details of the investigation, provided an assurance of confidentiality to the persons being surveyed, received the consent to participate, and conducted a survey using the 1:1 interview method.

This study investigated the characteristics of the participants who had frozen their gametes or were considering gamete freezing at the time of conducting the survey. Gamete freezing was defined as the case where the subject answered that they had ‘Already frozen gametes’ to the question, ‘Have you frozen or intend to freeze eggs or sperm for future pregnancy?’ The final number of subjects included in the analysis was 14,040 adults, excluding those with missing values or no responses.

### 2.2. Study Variables

#### 2.2.1. Demographic Characteristics

The demographic characteristics of the subjects were ascertained using data on gender, age, education levels, residential area, marital status, religion, economic activity, and household income. The participants were classified according to age, namely, 19–30, 31–35, 36–40, 41–45, and 46–49 years. They were divided into three categories based on educational levels: ≤high school, university graduate, ≥graduate school. In the case of the residential area, individuals in -dong were classified as urban, while those in -eup or -myeon were categorized as rural. The marital status of the participants was categorized as single, married, divorced/separated, or widowed. The participants were classified according to whether they were following a religion or not (yes or no, respectively) and based on their economic activity, they were classified as ‘employed’ or ‘unemployed’. They were also classified according to their monthly income into four categories: less than 3 million won, 3 to 4.99 million won, 5 to 6.99 million won, and 7 million won or more.

#### 2.2.2. Reproductive Health Characteristics

The data on reproductive health characteristics obtained from the participants included contraception, genital infections, number of pregnancies, number of births, infertility, experience with infertility treatment, and attitudes toward marriage and children. Based on the literature review [7,17,24], the health behavior characteristics obtained from the participants were modified to suit the purpose of this study. Contraception was defined as ‘yes’ if the subject had ever used contraception. Genital infections in both men and women can be defined based on symptoms. Genital infection was defined as menstrual symptoms or reproductive system symptoms in women. In men, it was defined as sexual dysfunction, prostatitis, and benign prostatic hyperplasia. The total number of pregnancies to date, including any current pregnancy, was classified as 0, 1, 2, or 3 or more. The number of births comprised the number of children born, including deceased children, and was classified as 0, 1, 2, or 3. Infertility was defined as an inability to achieve pregnancy after having sex with the current spouse without contraception for at least 1 year. Infertility treatment was determined based on participants’ experience with artificial insemination or in vitro fertilization.

#### 2.2.3. Attitude Towards Marriage and Children

The attitudes of the participants regarding marriage were examined through the question, ’What do you think about marriage?’ The responses were divided into three categories: ‘Must do’ (‘Must do’ or ‘It’s better to do it’), ‘It’s better not to do it’, and ‘I don’t know’ (‘It’s okay to do it or not to do it’ or ‘I don’t know’). The attitude towards children was examined through the question, ‘Do you think you should have your own children?’ The responses were divided into three categories: ‘Must have’ (‘Must have’ or ‘It’s better to have children than not to have children’), ‘It doesn’t matter if you don’t have children’, and ‘I don’t know’.

### 2.3. Data Analysis

The collected data were analyzed using the SPSS/WIN 23.0 program as follows: The general characteristics of the subjects and whether they had undergone gamete freezing were analyzed using descriptive statistics such as frequency and percentage. Fisher’s Exact *p*-value was used for categorical data and the *t*-test was used for continuous variables. In particular, Fisher’s Exact *p*-value was used to compare variables depending on whether gamete freezing was performed. All of the statistical significance levels were set to *p* < 0.05.

## 3. Results

### 3.1. Characteristics of the Subjects Based on Their Choice of Gamete Freezing

The analysis of the data from the 2021 survey revealed that 101 participants (0.7%) were either wanting to or had already frozen their gametes (gamete-freezing group) while 13,939 participants (99.3%) (non-gamete freezing or non-freezing group) did not intend to freeze their gametes. In terms of sociodemographic characteristics, there were significant differences between the gamete-freezing group and the non-freezing group with respect to variables such as age, education, marital status, and household income (Table 1).

In the gamete-freezing group, 4% were 19–30 years old, 25.7% were 31–35 years old, 40.6% were 36–40 years old, 22.8% were 41–45 years old, and 6.9% were 46–49 years old. In the non-gamete freezing group, 30.3% were 19–30 years old, 14.7% were 31–35 years old, 20.0% were 36–40 years old, 20.1% were 41–45 years old, and 14.9% were 46–49 years old (*p* < 0.001). The average age was 38.1 years old in the gamete-freezing group and 35.7 years old in the non-freezing group. In the gamete-freezing group, 38.6% were male and 61.4% were female. In total, 16.9% were high school graduates or less, 76.2% were university graduates, and 6.9% had completed graduate school or higher (*p* = 0.021). In the non-freezing group, 40.6% were male and 59.4% were female.

In the gamete-freezing group, 85.1% lived in cities and 14.9% lived in rural areas, while in the non-freezing group, 82.4% lived in cities and 17.6% lived in rural areas. The marital status data included 4.9% single, 94.1% married, and 1.0% divorced or widowed individuals in the gamete-freezing group, and 35.9% single, 60.4% married, and 3.7% divorced or widowed in the non-freezing group, which was significant (*p* < 0.001). In the gamete-freezing group, 78.2% did not belong to any religion and 21.8% belonged to a religion, and in the non-freezing group, 70.5% did not belong to any religion and 29.5% belonged to a religion. In the gamete-freezing group, 66.3% were employed and 33.7% were unemployed, and in the non-freezing group, 66.8% were employed and 33.2% were unemployed. Regarding household income, 7.9% of the subjects earned less than 3 million won, 21.8% earned 3 to 4.99 million won, 39.6% earned 5 to 6.99 million won, and 30.7% earned 7 to 7 million won or more in the gamete-freezing group, and 16.4% earned less than 3 million won, 30.3% earned 3 to 4.99 million won, 27.7% earned 5 to 6.99 million won, and 25.6% earned 7 to 7 million won or more in the non-freezing group (*p* = 0.037) (Figure 1).

### 3.2. Reproductive Health and Attitudes of the Participants Towards Marriage and Children

In the areas of reproductive health and attitudes towards marriage and children under the subheadings of genital infection and sexual intercourse experience, there was a significant difference between the gamete freezing and non-freezing groups (Table 2).

In the gamete-freezing group, 29.7% of the 101 subjects said they had experienced genital infection, while this was true of 19.0% of the non-freezing group (*p* = 0.006). When asked about contraception, 80.2% of the gamete-freezing group and 85.4% of the non-freezing group said that they had used contraception. Regarding sexual intercourse experience, 100% of the gamete-freezing group and 88.6% of the non-freezing group said they had sexual intercourse experience (*p* < 0.001).

When asked about their attitudes regarding marriage, 46.5% of the gamete freezing group responded that they ‘must have’ marriage, 5% said that it was ‘better not to’, and 48.5% responded that they ‘don’t know’, while 45.2% of the non-freezing group responded that they ‘must have’ marriage, 5.5% said that it was ‘better not to’, and 49.3% responded that they ‘don’t know’. When asked about their attitudes towards children, 78.2% of the gamete freezing group responded that they ‘must have’ children, 20.8% said that they ‘don’t care’, and 1.0% responded that they ‘don’t know’, while 69.5% of the non-freezing group answered that they ‘must have’ children, 25.8% said that they ‘don’t care’, and 4.7% responded that they ‘don’t know’, showing an insignificant but meaningful result.

### 3.3. Obstetric History of Women Belonging to the Gamete-Freezing Group

We examined the obstetric history of 62 women who belonged to the gamete-freezing group (Table 3). As a result, there was a significant difference in infertility and the experience with infertility treatment. Among the 62 women, 61 women (98.4%) were married, 31 women (50%) had one pregnancy, and 32 women (51.6%) had experienced childbirth once. When asked about infertility, 57 women (91.9%) responded that they were infertile, and 5 women (8.1%) responded that they were not infertile (*p* = 0.016). A total of 53 women (85.5%) had some experience with infertility treatment and 9 women (14.5%) did not (*p* = 0.032).

## 4. Discussion

This study attempted to investigate the gamete freezing status and the reproductive health of adults aged 19–49 years using data from the 2021 National Family and Fertility Survey. Of the 14,040 subjects, 101 subjects had frozen their gametes, including 39 males (38.6%) and 62 females (61.4%). Since studies on gamete freezing have mainly been conducted on infertile women or patients with incurable diseases, this study, which was conducted on the representative sample of the general population covered by the above survey, could not be accurately compared with previous studies. Although the gamete freezing group comprised 0.7% of the study subjects, it is meaningful because it was conducted on the general population rather than a specific group. In addition, there is a limitation in that gamete freezing was defined as a single question without distinguishing between egg freezing and embryo freezing. Future studies will need to consider these issues and conduct various studies that consider the specific reasons for freezing, age at the time of freezing, freezing period and number, and cost.

As in some previous studies [5,24], the average age of the gamete-freezing group in this study was 38.1 years. However, the average age was 34.8 years in a study on breast cancer patients [22] and 32.8 years in a study on male patients with history of adolescent cancer [25]. Considering that the average age varies depending on the purpose of gamete freezing, various studies that are focused on the purpose will be necessary in the future. However, as marriages are delayed, the number of people freezing their eggs is expected to increase. Gamete freezing does not always entail a successful pregnancy and, considering the results of a study that found that the pregnancy rate after age 41 was found to be 23.3% [11], accurate information on gamete freezing and its impact on pregnancy and childbirth will be necessary [12].

There are many studies [5,12,19] that suggest that the educational level of people who preserve their gametes is high. In this study, many had graduated from college or higher. A study on gamete storage [17] stated that gamete storage would occur more frequently because women have higher education levels than men and educated women want educated men who can accompany them for childbirth. It is thought that this phenomenon will continue as women’s social advancement and self-determination rights are emphasized.

In the gamete-freezing group, it was found that more study subjects lived in cities than in rural areas. This seems to be because most hospitals with the technology to collect and store gametes are concentrated in large cities and most women of childbearing age live in cities. In this study, 94.1% of the subjects who had stored their gametes were married. In previous studies wherein the status of social gamete freezing was investigated, single people accounted for a large proportion of the research subjects, ranging from 49% [25] to 71% [12]. However, in this study, married people accounted for the majority as secondary data surveys, such as this one, are conducted on the entire population without considering social gamete freezing issues. In future studies, it may be necessary to address issues associated with social gamete freezing more elaborately. Based on this, the necessary information can be provided, or social policies can be established, for individuals who seek such assistance.

A total of 21.8% of the gamete freezing group responded that they belonged to a religion. Earlier research also suggests that if there is a high negative attitude of religions towards gamete freezing or in individuals who are more religious, the intention to preserve gametes is lower [9]. As most religions are struggling with the ethical values and religious standards that are consistently applied to the rapid development of medical technology, it seems that detailed research on this is also necessary in the future.

As in previous studies [5,12], 66.3% of respondents said they were economically active. In future studies, it would be meaningful to conduct a detailed survey by gender on employment type, field of work, etc. For women, freezing eggs requires a lot of time and money. Hence, it seems necessary to establish social systems such as the accessibility to treatment in hospitals and vacation systems that consider economically active subjects. One study reported [26] that income levels were unrelated to gamete freezing. However, in this study, the monthly income of the subjects was relatively high, with 71.3% earning ≥5 million won. In many countries, the procedures and freezing costs related to gamete freezing are high, but it has been mentioned that national support is necessary in considering equality in healthcare, etc. [12].

Considering earlier studies [27,28] that reported a correlation between genital infections and infertility, this study also investigated genital infections among the participants. A total of 29.7% of the study subjects responded that they had experienced symptoms of a genital infection. In a study conducted on 200 women of childbearing age aged 20–49 years [29], genital infection was defined based on a questionnaire on self-reported symptoms and the experience of treatment for vaginitis, and 95% of women experienced genital infection symptoms, while 9% had no sexual intercourse experience. In this study, female genital infections were defined as those with menstrual problems or reproductive system symptoms. In addition, there may be limitations in that the subjects were asked whether they had experienced genital infections at a time other than the time of the study or freezing gamete. But in future studies, specific items, such as the type of infection, time of occurrence, and treatment progress, should be considered.

In terms of the attitudes towards marriage, there were no significant results observed regarding the opinions on the necessity of marriage in the gamete-freezing or the non-freezing groups. This is believed to be a phenomenon similar to that related to the intentions of selective gamete freezing in an increasingly complex society [12]. In terms of attitudes toward bearing children, 78.2% of the gamete-freezing group said that having children is essential. This study showed the same results as studies that found that people who practice social gamete storage believe that they should have children [7,17].

Most women in the gamete freezing group (N = 62), were married. Nearly 50% of the participants went through only one pregnancy and 51.6% one childbirth. A total of 91.9% were diagnosed with infertility and 85.5% had some experience with infertility treatment. These results are because the study subjects were included up to 49 years of age, taking into account the delayed marriage and childbirth age and previous studies on gamete cryopreservation [9,30]. In addition, this study did not consider social gamete storage subjects and used secondary data from the general population. Future studies will need to include a more detailed evaluation of aspects such as marital status, reasons for gamete freezing, and specific obstetric history to tailor education and nursing intervention approaches to the target population.

There are studies [24,26] that show that people preserve gametes because they believe their fertility to be lower than that of others of the same age. However, this study used secondary data and could not confirm the specific reason for preserving gametes. In addition, only the status of gamete freezing was identified and the factors affecting it could not be identified. Future studies will need to specifically examine the reasons for gamete freezing by considering demographic and physical health variables. Gamete freezing will gradually increase for various social reasons, and studies should be conducted in anticipation of the problems that could occur. People should be provided with accurate information about gamete freezing [5]. Health equity that allows them to access these medical services should also be considered [31].

This study is meaningful in that it investigated the status of gamete freezing and reproductive health and attitudes towards marriage and children targeting a large sample of the general population rather than a specific population group. However, this study has the following limitations: first, the study defined gamete freezing based on a single question regarding whether gamete freezing was undertaken or whether the participants were interested in doing so. Regarding gamete freezing, it was not possible to confirm exactly whether it was egg freezing or embryo freezing. Second, this study did not include the reasons for gamete freezing and the variables that affect freezing. Since the study used secondary data, it was not possible to identify the significant factors affecting gamete freezing and only fragmentary research results could be presented. Third, since it was a survey study based on a questionnaire, the responses of the subjects may not be accurate. Considering these issues, future studies will need to consider various related variables to confirm the gamete freezing status and its influencing factors. In addition, these research results can be used as basic data for developing educational programs or policies related to gamete freezing.

## 5. Conclusions

This study investigated the status of gamete freezing in the representative Korean population and their sexual health using data from the 2021 National Family and Fertility Survey. Of the 14,040 study subjects, 101 (0.7%) had frozen their gametes or intended to freeze them. The average age of these subjects was 38.1 years, with 38.6% being male and 61.4% female. The gamete freezing group comprised mostly college graduates or higher, urban residents, and married. Of the subjects, 66.3% were economically active and their household income was relatively high. Genital infections were experienced by 29.7% of the subjects and 100% responded that they had sexual intercourse experiences. The obstetric history of women who had frozen their gametes showed that the most common number of pregnancies and childbirth was one, and 91.9% responded that they were infertile. The results of this study can be used as basic data for developing educational programs or policies for subjects who wish to freeze their gametes in the future.

## Figures and Tables

**Figure 1 healthcare-13-00210-f001:**
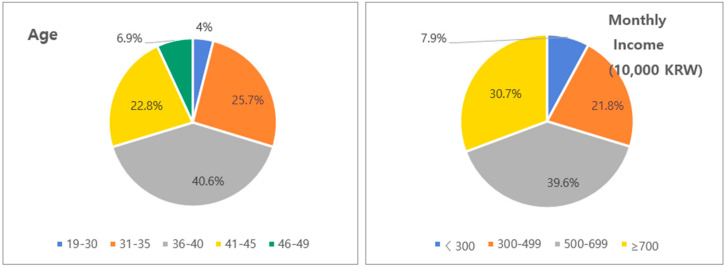
Age and monthly income of the gamete-freezing groups.

**Table 1 healthcare-13-00210-t001:** Characteristics of the gamete-freezing and non-freezing groups (N = 14,040).

Variables	Categories	YES (*n* = 101) N (%) or M ± SD	No (*n* = 13,939) N (%) or M ± SD	Fisher’s Exact *p*-Value
Age	19–30	4 (4)	4219 (30.3)	<0.001
	31–35	26 (25.7)	2057 (14.7)	
	36–40	41 (40.6)	2787 (20.0)	
	41–45	23 (22.8)	2798 (20.1)	
	46–49	7 (6.9)	2078 (14.9)	
	Total	38.11 ± 4.70	35.7 ± 8.62	
Sex	Male	39 (38.6)	5654 (40.6)	0.157
	Female	62 (61.4)	8285 (59.4)	
Education	≤High school	17 (16.9)	3666 (26.3)	0.021
	University	77 (76.2)	9518 (68.3)	
	≥Graduate school	7 (6.9)	755 (5.4)	
Residence	Urban	86 (85.1)	11,482 (82.4)	0.581
	Rural	15 (14.9)	2457 (17.6)	
Marital status	Single	5 (4.9)	4995 (35.9)	<0.001
	Married	95 (94.1)	8423 (60.4)	
	Divorce or widowed	1 (1.0)	521 (3.7)	
Religion	No	79 (78.2)	9824 (70.5)	0.115
	Yes	22 (21.8)	4115 (29.5)	
Economic Activity	No	34 (33.7)	4634 (33.2)	0.999
	Yes	67 (66.3)	9305 (66.8)	
Monthly Income (10,000 KRW)	<300	8 (7.9)	2286 (16.4)	0.037
	300–499	22 (21.8)	4224 (30.3)	
	500–699	40 (39.6)	3859 (27.7)	
	≥700	31 (30.7)	3570 (25.6)	

**Table 2 healthcare-13-00210-t002:** Reproductive health and attitudes towards marriage and children (N = 14,040).

Variables	Categories	YES (*n* = 101) *n* (%)	No (*n* = 13,939) *n* (%)	Fisher’s Exact *p*-Value
Genital infection	No	71 (70.3)	11,289 (81.0)	0.006
	Yes	30 (29.7)	2650 (19.0)	
Contraception	No	20 (19.8)	2038 (14.6)	0.168
	Yes	81 (80.2)	11,901 (85.4)	
Sexual intercourse experience	No	0 (0)	1593 (11.4)	<0.001
	Yes	101 (100)	12,346 (88.6)	
Attitudes towards marriage	I should get married	47 (46.5)	6301 (45.2)	0.951
	It’s better not to get married	5 (5.0)	767 (5.5)	
	I don’t know	49 (48.5)	6871 (49.3)	
Attitudes towards children	It must exist	79 (78.2)	9688 (69.5)	0.056
	I don’t care	21 (20.8)	3591 (25.8)	
	I don’t know	1 (1.0)	660 (4.7)	

**Table 3 healthcare-13-00210-t003:** Obstetric history of women belonging to the gamete-freezing group (N = 62).

Variables	Categories	N	%	Fisher’s Exact *p*-Value
Marital status	Single	1	1.6	0.862
	Married	61	98.4	
Pregnancy (N)	0	10	16.1	0.347
	1	31	50	
	2	12	19.4	
	≥3	9	14.5	
Childbirth (N)	0	20	32.3	0.231
	1	32	51.6	
	2	9	14.5	
	≥3	1	1.6	
Infertility	No	5	8.1	0.016
	Yes	57	91.9	
Experience with infertility treatments	No	9	14.5	0.032
	Yes	53	85.5	

## Data Availability

The data are available in a publicly accessible repository that does not issue DOIs. Publicly available datasets were analyzed in this study. These data can be found here: https://www.kihasa.re.kr/publish/report/research/view?searchText=%EA%B0%80%EC%A1%B1%EA%B3%BC&page=1&seq=46914 (accessed on 2 September 2024).

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
