# Peer review of "Study on the Status of Gamete Freezing and Reproductive Health of Korean Adults Aged 19–49 Years"

_healthcare, 2025, doi:10.3390/healthcare13030210_

Round 1
Reviewer 1 Report
Comments and Suggestions for Authors
This paper entitled “Study on the status of gamete freezing and reproductive health of Korean adults aged 19-49 years” relates to a descriptive study detailing the current status of Gamete freezing in Korean adults (not perception/attitude or potential for future freezing).
I recommend rejecting this paper for publication in its current form, or accept after major revisions. The topic of status/perception of gamete freezing is widely studied, and although there are no studies pertaining to the Korean population, the added benefit of this study with its current design is lacking.
Major:
- - Differentiating between egg freezing and embryo freezing: One of the major hurdles of this study is that it did not differentiate between the patients who did oocyte freezing versus those who did embryo or sperm freezing.
- - The marriage/couple status is confusing: 98% if the couples who were interviewed were already married. Did they do egg freezing? Or embryo freezing? Was it done prior to them being in a couple? All these info are crucial to understanding the perception of gamete freezing.
- - The percentage of patients undergoing gamete freezing is very low (0.7%), does this confer any bias in the results? Needs to be addressed in discussion
- - Many of the couples interviewed already had multiple pregnancies
- - The design/aim of the study seems flawed: If the aim of the study is to potentially use its data for the development of awareness programs or policies on gamete freezing, why did it include patients up to the age of 49? Why did it include patients that have already completed their childbearing potential?
Minor
- - Minor English language corrections
Author Response
We appreciate your critical review of our work and suggestions for improving the quality of our manuscript. Based on the comments, we made point-by-point responses to all of them, and associated modifications to the manuscript.
Thank you in advance for your time and attention.

Reviewer 2 Report
Comments and Suggestions for Authors
Dear Authors,
I want to extend my heartfelt thanks to you for your excellent work on this manuscript. It is a well-drafted, insightful study that addresses a critical topic in reproductive health. Your exploration of gamete freezing and its demographic and attitudinal aspects is both thorough and timely, contributing valuable findings to the field. P,ease find some comment in attached file

Author Response

(The authors gave the same response as above.)

Reviewer 3 Report
Comments and Suggestions for Authors
1. There is a random underline in this sentence: "They were divided into three categories based on educational levels"
2. Why did you use chi square for comparing continuous variables such as age? WHy not use a t-test?
3. "It was found that 85.1% of people living in cities preserved their gametes, which is higher than the 14.9% of people living in rural areas." - This sentence is backwards. 85.1% of people who froze their gametes lived in cities. The way you have written it implies most of the urban population has frozen gametes which is not true based on your data.
4. In your discussion of genital infection rates. As these diagnoses are self-reported, I wonder if the higher rates among those with frozen gametes are due to better access to reproductive care and the required workup prior to freezing gametes.
5. You repeat the stats presented in the results in both the discussion and conclusion section. I recommend rewriting these sections, removing repetitive percentiles.
Author Response

(The authors gave the same response as above.)
